# Environmental Regulation, Trade Comparative Advantage, and the Manufacturing Industry’s Green Transformation and Upgrading

**DOI:** 10.3390/ijerph17082823

**Published:** 2020-04-20

**Authors:** Mengqi Gong, Zhe You, Linting Wang, Jinhua Cheng

**Affiliations:** 1School of Law and Business, Wuhan Institute of Technology, Wuhan 430205, China; gongmq1992@163.com (M.G.); shenhaiyuwan@126.com (L.W.); 2Research Center for Resources and Environmental Economics, China University of Geosciences, Wuhan 430074, China; 3School of Economics and Management, China University of Geoscience, 388 Lumo Road, Wuhan 430074, China

**Keywords:** environmental regulation, trade comparative advantage, manufacturing, green transformation upgrade, green total factor productivity

## Abstract

This paper is the first to systematically review the theoretical mechanisms of environmental regulation and trade comparative advantage that affect the green transformation and upgrading of the manufacturing industry. On this basis, corresponding hypotheses are put forward. The non-radial and non-angle SBM (slacks-based measure) efficiency measurement model with undesirable outputs was used, combined with the use of the ML (green total factor productivity index) productivity index to measure green total factor productivity. Finally, the theoretical hypothesis was empirically tested using data from 27 manufacturing industries in China from 2005 to 2017. The results show the following: (1) There is a significant inverted U-shaped curve relationship between environmental regulation and the transformation of the manufacturing industry. In other words, as environmental regulation increases, its impact on the transformation and upgrading of the manufacturing industry is first promoted and then suppressed. (2) When there are no environmental regulations, the trade comparative advantage of the manufacturing industry is not conducive to industrial transformation. However, under the constraints of environmental regulations, the comparative advantage of trade will significantly promote the green transformation and upgrading of manufacturing. Therefore, in order to effectively promote transformation and upgrading of the manufacturing, this paper proposes the following policy recommendations: (1) The Chinese government should pay more attention to the impact of environmental regulation intensity on the transformation of manufacturing industries, further increase the intensity of environmental regulation within the reasonable range, and fully exert the positive effects of environmental regulation on the trade patterns and manufacturing industry transformation. (2) We should further optimize the structure of trade, realize the diversification of manufacturing import and export, and promote its transformation into high-end manufacturing. On this basis, green production technology in the manufacturing industry can be improved through the technology spillover effect. (3) Efforts should be made to improve the level of collaborative development between environmental regulation and trade patterns and to explore the transformation path of the manufacturing industry with the integration of environmental regulation and trade patterns.

## 1. Introduction

As environmental regulations have been conspicuously enhanced and as global economic integration has rapidly developed, the impact of environmental regulations on the comparative advantage of a country’s trade has attracted widespread attention among scholars [1,2]. Some scholars have pointed out that environmental regulations are not only a crucial factor that affects the comparative advantage of a country’s trade [3,4], but also a means by which the government can solve the “market failure” of environmental problems [5]. Increasing the intensity of environmental regulations can enforce a meticulous filter of industrial groups, thereby promoting adjustment of the industrial structure [6,7], which is conducive to industrial transformation and upgrading. However, because of the dual characteristics (the economic structure of modern industry and backward traditional agriculture) of the China’s economic development, environmental regulation policies cannot adopt a one-size-fits-all model. Therefore, in order to effectively improve environmental performance, China has introduced a series of guidelines and policies, including limiting and stopping production in some manufacturing industries. Although this policy, which is at the expense of economic benefits in exchange for environmental benefits, will have certain effects in the short term, it cannot be a long-term solution. As a large emerging country, China cannot abandon manufacturing in response to environmental pressures. For a long time, manufacturing will remain an important “ballast stone” for the national economy. The World Bank data show that since 2003, China’s manufacturing exports have accounted for more than 90% of merchandise exports. However, manufacturing is a highly polluting sector, which has led to a lot of energy consumption in China. Therefore, in order to eliminate the low-end lock-in status of the manufacturing industry, its transformation and upgrading are essential, and these processes depend on the implementation of regional environmental regulations to a certain extent. Changes in environmental regulations will inevitably have an impact on manufacturing production costs, which will significantly affect the comparative advantages of manufacturing trade. Comparative advantages will also affect the green transformation and upgrading of the industry through the learning effect. Therefore, it is important to discuss the relationship between environmental regulations, trade comparative advantage, and green transformation and upgrading of the manufacturing industry. Any ignorance of the impact of these aspects will lead to bias in the estimation results.

At this stage, the research on environmental regulation and comparative advantage has been ample, but the conclusions have differed. Some scholars have suggested that enhanced environmental regulation can significantly affect a country’s comparative advantage of trade [8,9]. Cole et al. [10] conducted an empirical study using the data of 41 industries in Japan and found that environmental regulations would affect the Japan’s comparative advantage of trade, thereby having an impact on import patterns. Millimet and Roy [11] utilized empirical analysis to examine the relationship between environmental efficiency and export trade, and they concluded that appropriate environmental policies were important sources of competitive advantage for industrial exporters. Ollivier [12] found that asymmetric environmental regulation may provide several countries with a comparative advantage in pollution-intensive industries and increase local pollution. Du and Li [13] posited that the scale effects of environmental regulation might reduce the comparative advantage of the China’s foreign trade and restrain the export performance of Chinese enterprises. Liu and Xie [14] revealed that environmental regulation has a promotion effect of approximately 2% on the export competitiveness of the China’s manufacturing industry. However, this effect is non-linear and displays a U-shaped tendency, indicating that certain prerequisites must be fulfilled to validate the Porter hypothesis. Other scholars have reached the opposite conclusion: they have claimed that the enhancement of environmental regulation will not have an impact on a country’s comparative advantage of trade. Harris et al. [15] argued that if an industry is heavily dependent on the country’s specific factor input, then environmental regulations will have no significant impact on the country’s comparative advantage in trade. On the basis of the HOV (Heckscher–Ohlin–Vanek) model, Cole and Elliott [16] used the sample data of 60 countries to analyze the influence that environmental regulations have on the comparative advantage of the trade of pollution-intensive products. They found that steel and chemical industries in capital-rich countries have comparative advantages, whereas in developed countries with abundant capital endowment, these industries do not shift, even if the intensity of environmental regulation is rising. Lu [17] analyzed the total samples of 95 countries and subsamples of 42 countries in 2005, and concluded that it is not advisable to reduce environmental regulations to promote an increase in the comparative advantage of pollution-intensive products. Li et al. [4] analyzed the impact of environmental regulation intensity on the comparative advantage of trade by using data from 30 industries in China, and they found that the China’s abundant labor factor endowment is the main reason that the comparative advantage of trade of industries is concentrated in clean industries.

Environmental regulation not only affects the comparative advantage of trade, but also contributes to research on green industrial transformation and upgrading. Ambec et al. [18] suggested that the enhancement of environmental regulation within a reasonable range is conducive to increasing the international competitiveness of the manufacturing industry. Yuan and Xie [19] performed an empirical analysis using the panel data of 30 provinces in China from 1999 to 2011, and they suggested that the enhancement of environmental regulation could effectively promote adjustment of the industrial structure; that is, it could increase the proportion of technology-intensive and knowledge-intensive industries. However, formal environmental regulation can have an impact on adjustment of the industrial structure by first inhibiting, then promoting, and, finally, inhibiting transformation and upgrading. Tong et al. [5] indicated that the impact of environmental regulation on industrial transformation and upgrading is J-shaped in the east, central, and west China. Yu and Sun [20] asserted that the enhancement of environmental regulation is conducive to promoting transformation and upgrading of the manufacturing industry. Yuan and Xiang [21] used the panel data of the China’s manufacturing industry from 2003 to 2014 to test the impact of environmental regulation on industrial innovation and green development; they stated that environmental regulation could not effectively promote the upgrading of the industrial structure. Yi et al. [22] used the panel data of 30 provinces in China from 2010 to 2017 and asserted that China’s environmental policy instruments do not provide a sufficient impetus for green technology innovation. Lin and Xu [23] estimated the impact of carbon tax on the green total factor productivity (GTFP) in China’s metallurgical industry. The results revealed that, during the research period, carbon tax had a limited effect on energy-saving and CO_2_ reduction in the metallurgical industry of China, and levying a carbon tax had a negative effect on the green total factor productivity. Shen et al. [24] analyzed the influence of environmental regulations on the environmental total factor productivity of industrial sectors. The results indicated that different types of environmental regulations exerted heterogeneous influences on the environmental total factor productivity in different industries.

In conclusion, it is easily observed that the research on the impact of environmental regulation on trade comparative advantage and green industrial transformation and upgrading has been ample in terms of the divisions and areas of research involved. However, there is little literature on the impact of trade comparative advantages on the manufacturing industry transformation and upgrading [25]. With the internal resource environment and the external pressure of international competition faced by the China’s manufacturing industry, different comparative advantages of trade are bound to have different influences on the China’s manufacturing industry. To analyze the effect of environmental regulation on the transformation and upgrading of the manufacturing, we must clarify the impact mechanism of environmental regulation on the comparative advantage of industry trade to provide a policy reference for China to accelerate transformation and upgrading of the manufacturing industry. This paper attempts to explore the following questions: Will the improvement of China’s environmental regulations affect transformation and upgrading of the manufacturing? Do environmental regulations determine China’s manufacturing comparative advantage? What impact will a comparative advantage have on the manufacturing transformation and upgrading? Although these individual questions have been answered in the existing literature, they have not been put into an analytical framework for systematic empirical analysis. Answering the above questions is conducive to accelerating transformation and upgrading of the China’s manufacturing industry. The innovation points of this paper are as follows: First, the direct and indirect mechanisms of environmental regulation that affect green transformation and upgrading of the manufacturing industry are systematically established. Second, environmental regulation, the comparative advantage of trade, and the manufacturing green transformation and upgrading are set in a unified research framework, thereby making up for the deficiency of existing research. In fact, except for the apparent learning effects that exist among environmental regulation, the comparative advantage of trade, and productivity, productivity determines the self-selection effect to a certain degree. Therefore, if the learning effect resulting from the comparative advantage is ignored as a mediator, the estimation results may include errors. Third, in the process of empirical analysis, the interaction terms of environmental regulation and the comparative advantage of trade are introduced to consider the impact of their interaction on transformation and upgrading of the manufacturing industry.

## 2. Theoretical Mechanism

### 2.1. The Direct Impact of Environmental Regulations on Green Transformation and Upgrading of the Manufacturing Industry

The influence of strengthened environmental regulations on transformation and upgrading of the manufacturing is mainly embodied in the following two aspects: one follows the cost theory: that is, from the static angle of analysis, given the technical level and consumer demand structure, with the related resource configuration remaining the same, environmental regulation enhancement can lead to a rise in enterprise production costs, thereby inhibiting enterprise competitiveness and eventually resulting in a dilemma situation between environmental regulation and transformation and upgrading of the manufacturing industry. The other one is the innovation compensation theory: that is, from the dynamic perspective of analysis, a reasonable intensity of environmental regulation is deemed to encourage enterprises to increase Research and development (R&D) input and optimize the allocation of resources. Thus, it will stimulate the enterprise “innovation offsets” effect and ultimately improve the efficiency of enterprise production. Hence, the enhancement of environmental regulation can bring about green production and efficiency, which is beneficial to transformation and upgrading of the manufacturing. Therefore, it can be inferred that the final impact of environmental regulation on the manufacturing transformation and upgrading depends on the size of the above two effects. If the production costs caused by environmental regulations increase significantly, then they will hinder the technological innovation of enterprises and are not conducive to transformation and upgrading of the manufacturing industry. Conversely, if the innovation compensation effect of environmental regulation is large, then it will stimulate the improvement of the green production technology of enterprises and promote transformation and upgrading of the manufacturing.

Therefore, it can be inferred that, as environmental regulation in the China’s manufacturing industry is enhanced, the costs that enterprises need to invest to meet environmental regulation standards will also rise. Under such conditions, some high-polluting enterprises are squeezed out, because they cannot meet environmental regulation standards, and some enterprises will carry out technological innovation in order to meet these standards. This will cause production factors to transfer from the high-pollution sphere to the low-pollution sphere. Then, clean production will gradually replace pollution-intensive industries, which will promote the industrial structure optimization and transformation and upgrading of the manufacturing industry. On the contrary, if the intensity of environmental regulation is too high, then enterprises cannot bear the increase in production costs resulting from environmental regulation. On the one hand, it will squeeze out most of the high-pollution industries. On the other hand, it will lead to a decline in the output of low-pollution sectors. In this case, although environmental quality will have improved, it will not be conducive to the formation of enterprise competitiveness, thus inhibiting transformation and upgrading of the manufacturing industry. On this basis, this paper proposes the following hypothesis:
**Hypothesis** **1.**The increase in environmental regulation will promote transformation and upgrading of the manufacturing industry to some extent. However, if the intensity of environmental regulation is high enough and is further increased, transformation and upgrading of the manufacturing industry will be inhibited to some degree. Therefore, it is believed that the relationship between environmental regulation and transformation and upgrading of the manufacturing industry presents an inverted U-shaped curve. That is, as environmental regulations increase, their impact on the transformation and upgrading of the manufacturing is first promoted and then suppressed.

### 2.2. Transmission Mechanism of Environmental Regulation, Trade Comparative Advantage, and Green Transformation and Upgrading of the Manufacturing Industry

According to classical and neoclassical trade theories, a country’s trade model depends on its comparative advantage. Stricter environmental regulations will lead to a rise in product prices, which will significantly increase the export costs for enterprises, thus reducing the comparative advantage of the industry in the process of export trade. Because enhanced environmental regulation significantly increases production costs for high-pollution industries, it is not conducive to the export of polluting industries; this promotes adjustment of the structure of industrial export. However, the effect of environmental regulation on low-pollution industries is relatively slight, and when the factors of production flow from high-pollution industries to low-pollution industries, low-pollution industries become more competitive, which is beneficial to the export of low-pollution industries and effectively promotes transformation and upgrading of the manufacturing. Therefore, when environmental regulation is low or does not exist, the environment, as a cheap factor input, will significantly promote a rise in the comparative advantage of manufacturing trade, which will significantly inhibit transformation and upgrading of the manufacturing industry. Additionally, with the improvement of environmental regulation, export costs for polluting industries will significantly rise, which is conducive to the improvement of the export structure of the manufacturing industry, and it will ultimately promote transformation and upgrading of the manufacturing industry. On this basis, this paper proposes the following hypotheses.
**Hypothesis** **2.**In the absence of environmental regulations, trade comparative advantage will significantly inhibit green transformation and upgrading of the manufacturing.
**Hypothesis** **3.**When there are environmental regulations, they will significantly inhibit the comparative advantage of trade in highly polluting industries, stimulate the industry to improve the export structure, and thus promote green transformation and upgrading of the manufacturing.

## 3. Measurement of the Manufacturing Transformation and Upgrading

### 3.1. Measurement Methods and Evaluation

Generally speaking, industrial upgrading refers to the transformation of the comparative advantage of an industry from the original resource endowment advantage to the relative high-end technology advantage depending on the technological progress. In order to bridge the gap with the manufacturing in developed countries and achieve manufacturing power, in October 2014, China put forward the concept of “made in China 2025” for the first time and, insisting on the basic policy of “innovation-driven, quality first, green development, structural optimization, and humanity-oriented,” definitively demanded that the energy consumption, material consumption, and pollutant emissions of the key industrial businesses and institutions reach the advanced level worldwide in 2025. In other words, efficient energy utilization and environmental protection are prerequisites of transformation and upgrading of the manufacturing industry. Therefore, to define the indicators of the manufacturing transformation and upgrading, it is necessary to comprehensively consider the increase in the expected output, the improvement of the production technology, and the decrease in the non-expected output. On this basis, in this paper, referring to the previous studies, the green total factor productivity (GTFP) is applied as the proxy variable of the manufacturing transformation and upgrading [26].

On the basis of the input and output data of 27 manufacturing industries in China, and referring to the method of Li et al. [27], the non-radial and non-angle SBM (slacks-based measure) model of the non-expected output is adopted to measure the GTFP of each industry under the assumption of a variable return to scale (VRS). According to the environmental technology function defined by Fare et al. [28], it is assumed that $X=\left(x_{\mathrm{ij}}\right)\in R_{n\times m}^{+}$ represents the input factor vector,${\;Y}^{g}=\left(y_{\mathrm{ij}}^{g}\right)\in R_{u\times m}^{+}$ represents the desirable output vector, and $Y^{b}=\left(y_{\mathrm{ij}}^{b}\right)\in R_{v\times m}^{+}$ represents the undesirable output vector. Then, environmental technology can be expressed as:(1)$$T\left(x\right)=\left\{\left(x,y^{g},y^{b}\right)|x\geq X\lambda ,y^{g}\leq Y^{g}\lambda ,y^{b}=Y^{b}\lambda ,\sum_{i=1}^{m}\lambda =1,\lambda \geq 0\right\}$$
where λ is the weight of the cross-sectional observations. If $\overset{m}{\underset{i=1}{\sum }}\lambda =1$, then the return on scale is variable (VRS). If λ ≥ 0 and the constraint condition that the sum of weights is equal to 1 is not considered, then the return to scale is constant (CRS). At this point, if each decision unit of the production system has three vectors, namely, input, desirable output, and undesirable output, then the model can be expressed as:(2)$$min\rho =\frac{1-\frac{1}{m}\sum_{i=1}^{m}\frac{s_{i}^{-}}{x_{i0}}}{1+\frac{1}{s+t}\left(\sum_{r=1}^{s}\frac{s_{r}^{g}}{y_{r0}^{g}}+\sum_{r=1}^{t}\frac{s_{p}^{b}}{y_{p0}^{b}}\right)}$$
$$s.t.\left(\mathrm{subject}\;\mathrm{to}\right)\;\;\sum_{j=1}^{n}\lambda_{j}x_{mj}+s_{i}^{-}=x_{0},\;m=1,2,\ldots ,m\;\sum_{j=1}^{n}\lambda_{j}y_{rj}^{g}-s_{r}^{g}=y_{0}^{g},\;r=1,2,\ldots ,s\;\sum_{j=1}^{n}\lambda_{j}y_{rj}^{b}-s_{p}^{b}=y_{0}^{b},\;p=1,2,\ldots ,s\;\lambda \geq 0,s_{i}^{-},s_{r}^{g},s_{p}^{b}\geq 0$$
where ρ is the environmental efficiency evaluation value.$\;s_{i}^{-},\;s_{p}^{b}$ refer to the redundancy of input and undesirable output, respectively, and $\;s_{r}^{g}\;\mathrm{refers}\;\mathrm{to}\;\mathrm{the}\;\mathrm{insufficient}\;\mathrm{desirable}\;\mathrm{output}.$
*m, r*, and *p* are the numbers of indicators for input, desirable output, and undesirable output, respectively. In order to calculate the green total factor productivity, this paper introduces the directional distance function. Referring to the method of Chung et al. [29], the ML (green total factor productivity index) can be calculated:(3)$$ML_{t}^{t+1}={\left\{\frac{1+{\vec{D}}_{0}^{t}\left(x_{t},y_{t}^{g},y_{t}^{b};g_{t}\right)}{1+{\vec{D}}_{0}^{t}\left(x_{t+1},y_{t+1}^{g},y_{t+1}^{b};g_{t+1}\right)}\times \frac{1+{\vec{D}}_{0}^{t+1}\left(x_{t},y_{t}^{g},y_{t}^{b};g_{t}\right)}{1+{\vec{D}}_{0}^{t+1}\left(x_{t+1},y_{t+1}^{g},y_{t+1}^{b};g_{t+1}\right)}\right\}}^{1/2}$$

The measured ML index can only reflect the growth rate of the green total factor productivity, but not the GTFP itself; therefore, referring to the practice of Qiu et al. [30], selecting 2004 for the base period, and assuming that the GTFP is 1, the 2005 GTFP equals the 2004 GTFP multiplied by the 2005 ML index, the 2006 GTFP equals the 2005 GTFP multiplied by the 2006 ML index, and so on for the GTFP in all years.

In terms of variable selection, capital (K), labor (L), and energy consumption (E) are taken as the input variables, the industrial sales output value (Y) is taken as the expected output, and emissions of carbon dioxide (CO_2_), sulfur dioxide (SO_2_), chemical oxygen demand (COD), wastewater, and solid waste are taken as the non-expected output. The capital input (K) is measured by the net fixed assets of each industry and converted into the fixed asset investment price index in 2004. In fact, the previous research has generally used the perpetual inventory method to account for capital stock, which probably has an uncertain result. Different selections of the depreciation rate and the capital base will affect the estimation result. Therefore, this article adopts the net value of the fixed assets as the proxy variable of the capital stock. The number of employees at the end of the year in industrial enterprises that are above the size of a sub-industry is selected to measure labor input (L). The data are from the China Labor Statistical Yearbook. The total energy consumption data by sector are used to measure energy consumption (E). The value of industrial sales (Y) is used to measure the size of the output. Carbon dioxide (CO_2_), sulfur dioxide (SO_2_), chemical oxygen demand (COD), and emissions of wastewater and solid wastes are the selected proxy variables of the expected output, and according to the accounting methods for carbon emissions from the list of greenhouse gas emissions in the guide released by the Intergovernmental Panel on Climate Change (IPCC), coal, oil, and natural gas are used as three kinds of fossil energy consumed by different industries to calculate the carbon emissions. The specific formula is ${\mathrm{CO}}_{2}=\sum_{i=1}^{3}{\mathrm{CO}}_{2,i}=\sum_{i=1}^{3}E_{i}\times {\mathrm{NCV}}_{i}\times {\mathrm{CEF}}_{i}\times {\mathrm{COF}}_{i}\times \left(\frac{44}{12}\right)$, where CO_2_ represents the estimate of carbon dioxide emissions, i represents coal, oil, and natural gas energy, E represents all kinds of consumption, NCV represents three net calorific values of the primary energy, CEF represents the carbon emission coefficient, and COF represents the carbide factor (the carbide factor of coal is set to 0.99, the carbide factors of oil and natural gas are set to 1). 44 and 12 represents the molecular weight of carbon dioxide and carbon, respectively. The other data are taken from the China Statistical Yearbook.

### 3.2. Principles of Industry Selection

Since 2011, the national economy industry classification standard has been revised; therefore, in order to guarantee integrity and consistency of the data, “rubber products” and “plastic products”, which were distinct before the 2011 revision, are merged into “rubber and plastic products”, and “auto manufacturing” and the “railway, shipbuilding, aerospace, and other transportation equipment manufacturing industry” are merged into the “transportation equipment manufacturing industry”. At the same time, the data involved are from the UN COMTRADE database (https://comtrade.un.org/), and the international trade industry classification standard of the United Nations (SITC Rev. 3) (the international trade industry classification standard of the United Nations) is different from the national economy industry classification standard in China. Therefore, referring to the industry classification standard, the statistical caliber is unified by merging the “agricultural and sideline products processing industry” and “food manufacturing” into “food processing and manufacturing”. Finally, the data of 27 manufacturing industries are included in the analysis.

## 4. Model Setup and Data Description

### 4.1. Model Setup

On the basis of availability of the data, this paper uses the panel data of 27 manufacturing industries in China from 2005 to 2017 for empirical analysis. In order to grasp the overall impact mechanism of environmental regulation and trade comparative advantage on transformation and upgrading of the manufacturing industry, the following dynamic panel model is established:(4)$${\text{GTFP}}_{\text{it}}=\alpha_{0}+\alpha_{1}{\text{GTFP}}_{\text{it}-1}+\alpha_{2}{\text{ERS}}_{\text{it}}+\alpha_{3}{\text{ERS}}_{\text{it}}^{2}+\alpha_{4}{\text{NEX}}_{\text{it}}+\alpha_{5}{\text{lnIC}}_{\text{it}}+\alpha_{6}{\text{lnRD}}_{\text{it}}+\;\alpha_{7}Z{\text{lnKL}}_{\text{it}}+\alpha_{8}{\text{lnES}}_{\text{it}}+\alpha_{9}{\text{lnEI}}_{\text{it}}+\alpha_{10}{\text{lnLP}}_{\text{it}}+\alpha_{11}{\text{ERS}}_{\text{it}}*{\text{NEX}}_{\text{it}}+\mu_{i}+v_{t}+\varepsilon_{it}$$
where GTFP_it_ represents the green total factor productivity as a measure of the industrial transformation and upgrading index, and ERS_it_ represents the environmental regulation. Considering that environmental regulation may have a non-linear relationship with the GTFP, this paper also includes the square of environmental regulation in the model. NEX_it_ represents the trade comparative advantage, lnIC_it_ represents the international competition, lnRD_it_ represents the R&D intensity, lnKL_it_ represents the conditions of factor endowments, lnES_it_ represents the energy structure, lnEI_it_ represents the energy intensity, and lnLP_it_ represents the labor productivity. In order to further investigate the impact of trade comparative advantages in different industries on transformation and upgrading of the manufacturing industry under environmental regulations, this paper also includes the interaction of environmental regulations and trade comparative advantages.

### 4.2. Data Description

Environmental regulation strength (ERS): Different from national and regional environmental regulation measurement methods, the environmental regulation strength of the manufacturing industry is related to not only the environmental regulation policies implemented by the countries and regions in which they apply, but also the willingness to implement environmental regulation. Therefore, to measure the intensity of environmental regulation in different industries, we should start from the attention paid by different industries to environmental protection. Previous studies have mainly used the following methods to measure the intensity of environmental regulation: (1) pollution control cost per unit of output used as a measure [31], (2) pollution emission intensity, that is, the amount of pollution per unit of output, used to measure the extent to which an economy complies with environmental regulations, (3) the income level directly adopted as the proxy variable of environmental regulation [19], (4) the number of environmental regulatory agencies that supervise enterprises’ emissions adopted for measurement [32], and (5) pollution emissions under environmental regulations used for measurement [33]. It is difficult to measure the intensity of environmental regulations by industry, because it is not only constrained by the current intensity of environmental regulations, but is also related to the industry’s willingness and ability to implement environmental regulations. Therefore, this paper refers to the method of Cole et al. [31] to measure the strength of environmental regulation; that is, the proportion of waste gas and wastewater governance costs to the industrial sales value of various industries used as a measure. This indicator can effectively reflect the degree of constraint of environmental regulations and the industry’s willingness to adhere to environmental governance.

Trade comparative advantage (TCA): Balassa [34] thought that the revealed comparative advantage (RCA) could better measure a country’s trade comparative advantage. The computation formula is as follows: $\mathrm{RCA}=(X_{\mathrm{it}}/\sum_{i}X_{\mathrm{it}})/\left(X_{\mathrm{iwt}}/\sum_{i}X_{\mathrm{iwt}}\right)$, where X_it_ represents the exports of product i in year t, and X_iwt_ represents the total export of product i around the world in year t. An RCA value of less than 1 suggests that the country’s product i does not have a comparative advantage. If the value is more than 1, then the country has an obvious comparative advantage in i products. However, the result of the index for measuring a country’s trade comparative advantage only considers the export trade, ignoring the effect of import on the comparative advantage of trade. This can lead to estimation bias in the results. Therefore, this paper uses the net export index (NEX) to measure the trade comparative advantages and uses the Michaely index (MIC) to test the robustness of the model. The calculation formula of the net export index (NEX) is ${\mathrm{NEX}}_{\mathrm{it}}=\left({\mathrm{EX}}_{\mathrm{it}}-{\mathrm{IM}}_{\mathrm{it}}\right)/\left({\mathrm{EX}}_{\mathrm{it}}+{\mathrm{IM}}_{\mathrm{it}}\right)$, where EX_it_ and IM_it_ represent the export and import of product i in year t, respectively, with ${\mathrm{NEX}}_{\mathrm{it}}\in \left[-1,\;1\right]$. When the value is −1, product i is only imported and not exported. When the value is 1, product i is only exported and not imported. The calculation formula of the Michaely index (MIC) can be expressed as ${\mathrm{MIC}}_{\mathrm{it}}=\left({\mathrm{EX}}_{\mathrm{it}}/\sum_{i}{\mathrm{EX}}_{\mathrm{it}}\right)-\left({\mathrm{IM}}_{\mathrm{it}}/\sum_{i}{\mathrm{IM}}_{\mathrm{it}}\right).$ The import and export data are from the UN COMTRADE database.

Other control variables are as follows: (1) International competition (IC) measured by the proportion of total import and export by industry to the industrial output value of industrial enterprises above a designated size. (2) Research and development (R&D) investment intensity. In general, the higher the intensity of an industry’s R&D investment, the more conducive it is to the industry’s transformation and upgrading. Hence, the internal R&D investment per employee in classified industries is adopted as a measure. (3) Factor endowment (KL) measured by the capital–labor ratio, (4) energy structure (ES) measured by the proportion of coal consumption to the total amount of energy consumption, and (5) energy intensity (EI) measured by the proportion of total energy consumption to industrial sales. (6) Labor productivity (LP): this paper uses the ratio of the industrial sales output and employees as a measure.

The selection results of all variable indexes are shown in Table 1.

## 5. Regression Results

### 5.1. Regression Analysis Based on Green Total Factor Productivity

According to the dynamic panel regression equation (Equation (4)) set above, the method of system GMM (generalized method of moments) is used to carry out regression to Equation (4). The specific results are shown in Table 2. In order to describe the influence of addition of the control variables on the regression results, the model is regressed by adding the control variables step-by-step.

First, from the perspective of the impact of environmental regulations on the GTFP, the coefficient symbols of primary and secondary terms of environmental regulations are positive and negative, respectively, and most of them pass the significance test. It can be found that the relationship between environmental regulation and the GTFP is an inverted U-shaped curve. To some extent, this result confirms hypothesis 1. This shows that there is a significant non-linear relationship between environmental regulations and the GTFP [27,35]. With the strengthening of environmental regulations, the GTFP of the China’s manufacturing industry first increases and then decreases. In other words, environmental regulations must be controlled within a reasonable range to be conducive to the promotion of the GTFP in the China’s manufacturing industry. Once the level of environmental regulation exceeds the inflection point of the inverted U-shaped curve, it will significantly increase the costs for the manufacturing industry, then inhibit the GTFP, and ultimately, inhibit transformation and upgrading of the China’s manufacturing industry. Second, from the perspective of the impact of trade comparative advantage on the GTFP, it will significantly inhibit the improvement of the GTFP in the manufacturing. The reason is that without considering environmental regulations, the environment as a cheap factor input will significantly promote a reduction in the China’s manufacturing production costs while promoting the trade comparative advantage of the manufacturing. At this time, because of its comparative advantages, China will specialize in the production of such products. However, the manufacturing industry is a pollution-intensive industry, which makes China a “pollution haven” and is unfavorable for the improvement of its GTFP, thereby inhibiting transformation and upgrading of the manufacturing. This result confirms hypothesis 2. Third, from the perspective of the interaction between environmental regulation and trade comparative advantage, environmental regulation will significantly promote the improvement of the GTFP and pass the significance test at the significance level of at least 1%. This shows that when there are environmental regulations, some highly polluting enterprises must increase investment in order to meet environmental regulatory standards. At this time, the production costs for the enterprises rise, which will change the trade comparative advantage of the industry to a certain extent. Then, some high-pollution companies will withdraw from the market, because they will fail to meet environmental regulatory standards, and some companies will alleviate the increase in costs caused by stricter environmental regulations by promoting innovation in the production technology. This is conducive to the growth of the manufacturing GTFP and promotes transformation and upgrading of the manufacturing. This finding could be interpreted through the “pollution paradise hypothesis” [36,37]. Differences in environmental regulation intensity among countries have changed comparative advantages of their industrial sectors, thus encouraging industrial sectors with high environmental regulation intensity to transfer their polluting industries to countries with low environmental regulation intensity and ultimately promoting green industrial transformation and upgrading. So far, hypothesis 3 has been proven.

Finally, from the perspective of the other control variables, the following results are found: (1) For every percentage point increase in the level of international competition, the GTFP will increase by 0.0005–0.0010 percentage points and pass the significance test at the significance level of at least 10%, indicating that the level of international competition will generate a positive technology spillover effect to the GTFP industry, which is conducive to transformation and upgrading of the manufacturing industry. (2) The increase in R&D spending will significantly inhibit the GTFP. The reasons for this phenomenon may be related to improper R&D structure at the present stage, and because the utilization rate of R&D results and their practical application effects are poor. Another possible explanation is that a non-linear relationship emerges between R&D and the GTFP, and R&D has a certain time-lag effect, while the present R&D input is still in the primary stage of the Chinese manufacturing industry, so its positive influence on the GTFP has not emerged. When these conditions exist, the increase in the current R&D investment will not effectively promote the improvement of the GTFP, but, on the contrary, will inhibit the growth of productivity because of the excessive use of funds. (3) Factor intensity will significantly promote the improvement of the GTFP, indicating that the higher the degree of capital intensity, the more conducive it is to promoting technological innovation. (4) The influence of the energy structure on the GTFP is significantly positive, but it does not pass the significance test. (5) The influence of energy intensity on the GTFP is significantly negative, and it has passed the significance test at the significance level of at least 5%, indicating that the energy utilization efficiency of the Chinese manufacturing industry still needs to be improved at this stage. (6) Labor productivity will significantly inhibit the GTFP, which may be related to a need to improve China’s labor productivity.

### 5.2. Robustness Test

#### 5.2.1. The Estimated Result of Replacing the Explanatory Variable

Above, the GTFP is taken as the proxy variable of the manufacturing transformation and upgrading. The influence of environmental regulation and trade comparative advantage on the manufacturing transformation and upgrading is investigated. It should be noted that the GTFP, as the input, expected output, and unexpected output of the comprehensive index, furthers understanding of the industry’s technological innovation effect overall, but it cannot effectively consider the technical efficiency and the technical progress effect caused by environmental regulation and trade comparative advantage in the process of improving the GTFP. Therefore, this part further takes the technical efficiency and the technical progress index as an explanatory variable to analyze the influence of environmental regulation and trade comparative advantage regression on transformation and upgrading of the manufacturing industry. The method adopted for the regression, namely, the gradual increase in the control variable, is also used here. The specific results are shown in Table 3, which shows that the coefficient of environmental regulation is positively statistically significant, and the quadratic term coefficient of environmental regulation is negative at the significance level of at least 1%. This illustrates that the inverted U-shaped curve relationship between environmental regulation and the green transformation and upgrading of the manufacturing is still applicable. Meanwhile, the influence of trade comparative advantage on technical efficiency and the technological progress is significantly negative, and it passes the significance test at the significance level of 1%, indicating that the trade comparative advantage of the China’s manufacturing industry will increase the negative influence on technical efficiency and technological progress at the present stage, which is also consistent with the previous conclusion. From the angles of environmental regulation and comparative advantages of trade interaction, it can be found that, under the constraint of environmental regulation, trade comparative advantage will significantly promote the improvement of technical efficiency and technical progress, and the result is highly robust. Because of the rising production costs that result from environmental regulation, companies that fail to meet the requirements of local environmental regulation exit the market. At the same time, these costs stimulate some enterprises to innovate the technology. To some extent, it will promote the trade comparative advantage of cleaning products and promote transformation and upgrading of the manufacturing industry.

#### 5.2.2. The Estimated Results of Replacing the Core Explanatory Variables

The Michaely index (MIC) is adopted to replace the trade comparative advantage index, and referring to the practice of Zhang et al. [38], the ratio of the investment for environmental pollution governance to the main business costs is used to measure the intensity of environmental regulation (ERS1). The difference of the GMM and the system GMM were adopted to test the model, and the regression results are shown in Table 4, which shows that the main variable regression coefficient symbol and the significance level are consistent with the data above, illustrating that for the metrological test result of the model, the article above has good robustness.

### 5.3. Further Analysis

Because industries are intensive in different factors, there is heterogeneity in the demand for energy. This heterogeneity lies in the fact that environmental regulation has different impacts on the trade comparative advantage and the GTFP of industries that are intensive in different elements. Therefore, this paper further adopts the China’s manufacturing industry’s overall average as the division standard of factor endowments, and the 27 manufacturing sectors are divided into capital-intensive industries and labor-intensive industries. When the factor endowment is lower than the overall average, the industry is divided into labor-intensive industries, and, on the contrary, it is divided into capital-intensive industries when the factor endowment is larger than the overall average. The regression results are shown in Table 5. First, it can be observed that in both capital-intensive industries and labor-intensive industries, environmental regulation and the GTFP have an inverted U-shaped curve relationship. However, the influence of environmental regulation on the GTFP degree of capital-intensive industries is more distinguished. The main reason lies in the fact that capital-intensive industries are more sensitive to changes in environmental regulation, and the degree of dependence on environmental factors in the process of production is stronger. Second, the trade comparative advantage of both capital-intensive and labor-intensive industries has a negative impact on the GTFP. However, excessive dependence on the environment in capital-intensive industries, as well as the low technology and low value-added characteristics of labor-intensive industries, leads to their disadvantages in industrial transformation and upgrading. Finally, under the constraint of environmental regulation, the industry’s trade comparative advantage will promote the growth of the GTFP. The benefit of improvement is more obvious in labor-intensive industries. The main reason is that labor-intensive industries are less affected by environmental regulation than capital-intensive industries. Stricter environmental regulation stimulates the technological innovation of enterprises. For the capital-intensive industries with higher energy intensity, technological innovation needs more investment; therefore, the promoting effect is poorer than that in labor-intensive industries.

## 6. Conclusions and Policy Proposal

First, the direct effects of environmental regulation on the green transformation and upgrading of the manufacturing industry and the transmission mechanism between environmental regulation, trade comparative advantage, and the green transformation and upgrading of the manufacturing industry were systematically analyzed. Second, from the specific aspect of measuring the green transformation and upgrading of the manufacturing, we found that, on the whole, the manufacturing GTFP is rising. Finally, taking the data of 27 manufacturing industries in China as the research object, below, this paper reports the results of empirical tests, using the data from 2005 to 2017 to determine the impact of environmental regulation and trade comparative advantage on the green transformation and upgrading of the manufacturing industry, as well as the overall mechanism of its effect. The empirical results showed the following:

First of all, a significant inverted U-shaped curve relationship between environmental regulation and the GTFP was evident, which indicates that increasing the intensity of environmental regulation within a reasonable scope is beneficial for the promotion of transformation and upgrading of the manufacturing. However, if environmental regulation is enhanced blindly, at the turning point of the inverted U-shaped curve, the production efficiency of manufacturing enterprises will be inhibited because of the increase in production costs, which is not conducive to enhancing enterprise competitiveness. Meanwhile, it can also inhibit the green transformation and upgrading of the manufacturing industry. This conclusion is consistent with hypothesis 1 proposed in the theoretical part of this paper.

Second, at the present stage, the trade comparative advantage of the China’s manufacturing industry will significantly inhibit the improvement of the GTFP, which is not conducive to the green transformation and upgrading of the manufacturing industry. In this paper, the net export index (NEX) and the Michaely index (MIC) were used to measure the trade comparative advantage of the manufacturing industry. It can be observed that, except for some industries, the China’s manufacturing industry does have a significant comparative advantage in participating in the international trade. Further, comparative advantage theory points out that a country or a region that prefers to professionally produce products has a comparative advantage. Manufacturing is associated with pollution-intensive industries, and when they professionally produce this kind of product, on the one hand, it can lead to a significant reduction in the quality of the national environment, but on the other hand, it is also not conducive to the adjustment of the industrial structure. Eventually, it will curb the green transformation and upgrading of the manufacturing industry. This conclusion is consistent with hypothesis 2 proposed in the theoretical part of this paper. This result shows that when environmental regulations are not considered, the highly polluting manufacturing industry’s comparative advantage of trade will significantly inhibit its transformation and upgrading.

Third, according to the environmental regulation and trade comparative interaction coefficient, environmental regulation will significantly promote the improvement of the manufacturing GTFP, which shows that the increase in environmental regulation will affect the manufacturing industry’s trade comparative advantage to a certain extent. Under the constraint of environmental regulation, an initial comparative advantage may be lost, because stricter environmental regulations lead to an increase in production costs, which will change the industry’s trade structure to some extent. To be specific, on the one hand, a stricter environmental regulation will stimulate enterprises to invest in R&D to improve the production technology by increasing production costs, promoting them to move to the production frontier for green production. On the other hand, when environmental regulation reaches a certain level, the few enterprises that fail to meet environmental regulation standards will exit the market, the number of companies constrained will be significantly reduced, and the regulated industry market concentration will be increased. The surviving companies will have stronger competitiveness and tend to focus more on the technological progress, which is conducive to improving the industry’s production technology, thereby promoting transformation and upgrading of the green manufacturing in the end. This conclusion is consistent with hypothesis 3 proposed in the theoretical part of this paper.

In light of the state of the China’s manufacturing (big, but not strong, stagnated technological innovation, severe overcapacity caused by falling demand, effective supply lagging behind the consumption structure upgrade, industries’ comparative advantages of trade, and seriously influenced environmental quality), difficulties are faced in the green transformation and upgrading of China’s manufacturing industry. Therefore, application of environmental regulations to influence the trade comparative advantage of industries and the implementation of the “filtered wash” for industries have gradually become the social consensus. On the basis of the above conclusions combined with the trade comparative advantages of the China’s manufacturing industry and the difficulties it faces, this paper puts forward the following suggestions.

First, the Chinese government should pay more attention to the impact of environmental regulation intensity on the green transformation and upgrading of the manufacturing industry, further improve environmental regulation intensity within a reasonable range, and give full play to the positive effect of environmental regulation on trade comparative advantage and the green transformation and upgrading of the manufacturing industry. According to the conclusions of this research, when environmental regulation of the manufacturing is at the left of the inverted U-shaped curve, it is beneficial for the green transformation and upgrading of the manufacturing. Therefore, before the manufacturing of the environmental regulation intensity across the inverted U-shaped curve reaches the turning point, the government should aim to formulate a reasonable environmental regulation range according to the realistic characteristics of the industry, instead of blindly increasing the intensity of environmental regulation. Within a reasonable range, the government should revise the environmental regulation policy to avoid conditioning in a constant, static environmental regulation standard, which forms a reversed transmission mechanism of the enterprise, leading to the transition of the enterprise to intensive green production.

Second, the import and export trade structure of the manufacturing industry should be optimized. From the perspective of the regression results in this paper, although the trade comparative advantage of the China’s manufacturing industry has a negative effect on the industrial transformation and upgrading, this does not mean that manufacturing exports should be eliminated. The key at this stage is to adjust the export structure, change the previous trade comparative advantage, strengthen enterprises in export in the process of acquiring self-learning ability, and through the technology spillover effect, promote green production technology in the manufacturing industry. Particularly, in light of the United States’ increasing tendency toward the “conservative environmental protection,” to avoid the re-emergence of the international environmental responsibility, China should clearly realize its gap with the developed countries in the aspect of its resource-bearing capacity, strive to promote development of the green manufacturing idea, and focus on participating in the international trade in the process of developing long-term interests, rather than having the China’s economy once again return to the “gray economy” era because of short-term interests. All this will finally promote improvement of the import and export structure, implementation of enterprise technology, and participation in the high end of the global value chain division of labor to obtain more market power.

Finally, efforts should be made to improve the level of coordinated development between environmental regulation and trade comparative advantage, and the path of the manufacturing transformation and upgrading should be explored while integrating environmental regulation and trade comparative advantage. From the perspective of the regression results of its influence on the GTFP, environmental regulation under the restriction of trade comparative advantage will significantly promote transformation and upgrading of the manufacturing industry. This suggests that the current integration of environmental regulation and trade comparative advantage has a certain efficiency. However, transformation and upgrading of the manufacturing is a complex process that relies on a combination of various factors. Because a stricter environmental regulation is conducive to the improvement of the trade comparative advantage, this process will significantly promote transformation and upgrading of the manufacturing. Therefore, by enhancing the strength of environmental regulation in a reasonable range, the formation of trade comparative advantage and environmental regulation can be made to promote transformation and upgrading of the manufacturing industry and, ultimately, the green transformation and upgrading of the manufacturing industry.

## Figures and Tables

**Table 1 ijerph-17-02823-t001:** The selection results of variable indexes.

Type	Variable	Name
Explained variable	GTFP_it_	Green total factor productivity
Explanatory variables	ERS_it_	Environmental regulation strength
	NEX_it_	Trade comparative advantage
Control variable	lnIC_it_	International competition
	lnRD_it_	Research and development investment intensity
	lnKL_it_	Factor endowment
	lnES_it_	Energy structure
	lnEI_it_	Energy intensity
	lnLP_it_	Labor productivity

**Table 2 ijerph-17-02823-t002:** Estimation results.

Variable	(1)	(2)	(3)	(4)	(5)
GTFP_it-1_	1.1582 *** (0.0062)	1.1627 *** (0.0059)	1.1728 *** (0.0134)	1.1312 *** (0.0257)	1.1287 *** (0.0251)
ERS_it_	0.5569 ** (0.2256)	1.3060 *** (0.2717)	1.6005 *** (0.2854)	1.3675 ** (0.6428)	2.5052 ** (1.1966)
ERS^2^_it_	−1.3931 *** (0.2338)	−2.1921 *** (0.2435)	−2.8475 *** (0.2631)	−2.3545 *** (0.6095)	−3.4176 *** (1.1157)
NEX_it_	−0.1377 *** (0.0163)	−0.1590 *** (0.0342)	−0.3187 *** (0.0510)	−0.2709 ** (0.1159)	−0.3090 *** (0.1055)
ERS_it_×NEX_it_	1.0739 *** (0.0998)	1.3097 *** (0.2124)	1.9563 *** (0.2931)	1.7643 *** (0.6239)	2.0395 *** (0.6230)
lnIC_it_		0.0005 *** (0.0001)	0.0005 ** (0.0002)	0.0009 ** (0.0004)	0.0010 * (0.0005)
lnRD_it_			−0.0342 * (0.0191)	−0.0708 ** (0.0310)	−0.0425 (0.0560)
lnKL_it_				0.0030 *** (0.0009)	0.0040 *** (0.0015)
lnES_it_				0.0007 (0.0009)	0.0011 (0.0014)
lnEI_it_					−0.3557 ** (0.1541)
lnLP_it_					−0.0007 ** (0.0004)
_cons	−0.0885 *** (0.3221)	−0.1747 *** (0.0478)	−0.1595 ** (0.0658)	−0.1904 (0.1276)	−0.1442 (0.1238)
AR(1)	−1.11 [0.265]	−1.12 [0.264]	−1.11 [0.265]	−1.11 [0.266]	−1.11 [0.265]
AR(2)	−1.26 [0.209]	−1.25 [0.212]	−1.26 [0.206]	−1.23 [0.219]	−1.21 [0.228]
Sargan	9.85 [0.997]	9.63 [0.996]	8.10 [0.991]	9.20 [0.988]	7.48 [0.991]

Note: *, **, and *** mean significant at the levels of 10%, 5%, and 1%, respectively; the standard error of the estimated coefficient is shown in parentheses, and the *p*-value of the statistic is shown in square brackets.

**Table 3 ijerph-17-02823-t003:** The robustness test results after replacing the explanatory variables.

Variable	Technical Efficiency (TE)		Technical Progress (TP)	
	(1)	(2)	(3)	(4)
TE_it-1_ or TP_it-1_	1.2491 *** (0.0326)	0.7426 *** (0.0879)	0.3717 *** (0.0148)	0.1989 *** (0.0324)
ERS_it_	0.1802 *** (0.0333)	0.8901 *** (0.1767)	0.3103 *** (0.0850)	2.1225 *** (0.5529)
ERS^2^_it_	−0.2741 *** (0.0305)	−0.5169 *** (0.1437)	−0.5750 *** (0.1077)	−2.2927 *** (0.4835)
NEX_it_	−0.0520 *** (0.0043)	−0.0821 *** (0.0104)	−0.0802 *** (0.0229)	−0.1459 *** (0.0319)
ERS_it_×NEX_it_	0.3246 *** (0.0272)	0.7644 *** (0.0682)	0.4440 *** (0.1094)	1.2746 *** (0.2560)
Control variable	No	Yes	No	Yes
_cons	−0.2577 *** (0.0320)	0.2749 *** (0.0870)	0.7726 *** (0.0158)	0.8920 *** (0.0521)
AR(1)	−2.04 [0.042]	−1.76 [0.078]	−0.85 [0.396]	−0.31 [0.159]
AR(2)	1.33 [0.182]	1.27 [0.203]	1.26 [0.208]	1.23 [0.220]
Sargan	30.91 [0.192]	15.61 [0.271]	48.54 [0.411]	17.92 [0.267]

Note: *** mean significant at the levels of 1%, respectively; the standard error of the estimated coefficient is shown in parentheses, and the *p*-value of the statistic is shown in square brackets.

**Table 4 ijerph-17-02823-t004:** The robustness test results after the explanatory variables are replaced.

Variable	DIF-GMM	SYS-GMM
GTFP_it-1_	1.2793 *** (0.0421)	1.1279 *** (0.0215)
ERS1_it_	4.6539 *** (0.5740)	2.7701 *** (0.1428)
ERS1^2^_it_	−3.9930 *** (0.7585)	−2.7176 *** (0.1676)
MIC_it_	−9.8398 ** (4.0689)	−5.5430 *** (0.6831)
ERS_it_×MIC_it_	4.3766 (11.5683)	10.3627 *** (3.6807)
Controlled variables	Yes	Yes
_cons		−0.6927 (0.4369)
AR(1)	−1.15 [0.249]	−1.12 [0.262]
AR(2)	−1.11 [0.268]	−1.13 [0.258]
Sargan	8.27 [0.998]	9.86 [0.999]

Note: DIF-GMM and SYS-GMM mean the difference of the GMM and the system GMM; **, and *** mean significant at the levels of 10%, 5%, and 1%, respectively; the standard error of the estimated coefficient is shown in parentheses, and the *p*-value of the statistic is shown in square brackets. MIC: Michaely index.

**Table 5 ijerph-17-02823-t005:** Estimated results grouped by factor endowment.

Variable	Capital-Intensive		Labor-Intensive	
	(1)	(2)	(3)	(4)
GTFP_it-1_	1.1684 *** (0.0256)	1.2914 *** (0.1321)	0.6385 *** (0.0866)	1.6702 ** (0.8414)
ERS_it_	2.1115 *** (0.6238)	8.5228 *** (2.1467)	0.6479 *** (0.1362)	4.1256 *** (1.5643)
ERS^2^_it_	−1.8501 *** (0.5646)	−7.6851 *** (1.8063)	−1.2422 *** (0.3986)	−7.2071 *** (1.1631)
NEX_it_	−1.5915 ** (0.7330)	−0.7749 *** (0.2371)	−0.0702 *** (0.0107)	−0.5780 ** (0.2664)
ERS_it_×NEX_it_	1.6373 * (0.9623)	1.5820 * (0.9121)	1.7484 *** (0.0796)	6.4427 ** (2.7870)
Controlled variables	No	Yes	No	Yes
_cons	−0.5955 *** (0.1865)	−2.7392 (1.9404)	0.4045 *** (0.0826)	1.5864 (1.2953)
AR(1)	−1.77 [0.077]	−1.62 [0.105]	−2.97 [0.003]	−1.00 [0.318]
AR(2)	−1.19 [0.233]	−1.02 [0.307]	0.71 [0.478]	−0.04 [0.968]
Sargan	16.78 [0.115]	7.81 [0.167]	10.47 [0.163]	10.86 [0.369]

Note: *, **, and *** mean significant at the levels of 10%, 5%, and 1%, respectively; the standard error of the estimated coefficient is shown in parentheses, and the *p*-value of the statistic is shown in square brackets.

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
