# Peer review of "Environmental Regulation, Trade Comparative Advantage, and the Manufacturing Industry’s Green Transformation and Upgrading"

_ijerph, 2020, doi:10.3390/ijerph17082823_

Round 1
Reviewer 1 Report
The objective of the paper in the writers' own words: to explore the impact of China's comparative advantage of trade on the green transformation and upgrading of manufacturing industry under the background of strengthened environmental regulations, and provide suggestions for China to formulate appropriate environmental regulations and trade policies to promote the green transformation and upgrading of manufacturing industry” which is investigated in the regression analysis. This is an important and relevant question considering the very significant share and role of China in the world economy and manufacturing. However, this high importance is not highlighted with some relevant figures in the text.
As it is clear from the sentence stating the manuscript’s objective above and as from the rest of the text, not much attention is paid to the use of language. The potential of the manuscript could much better be released, if it is revised and edited thoroughly before any resubmission. The review was difficult due to the wordy way the manuscript was written. Whole text must be in shorter, clearer sentences and with less repetition as it was in the parts on regression analysis and data (Sections 4 and 5). To illustrate my point I attach more particular comments by section at the end referring the line number. Off course, reading the whole text at this level of completion this critically would not be possible as a reviewer.
I think this paper is still in a developmental phase and there are central points that must be addressed.
- The literature review does not formulate the problem, its significance and the literature gap. It does not highlight the originality of the research. A brief Google scholar search showed me that there is a considerable body of recent literature on green industrial transformation in China. Citing these articles and discussing the contribution made with regards to gaps in this literature, justification of theory and quantitative methods used, (e.g. the advantage of methods use for analysis in the existing literature, why Cole et at.’ s method is used for environmental regulation strength measurement but not another method) would significantly improve the research formulation.
- There is a theory section but no linkage between theory and discussions. Discussions are not fully validated with the findings. In section 6, if there are any the theoretical and practical contributions of this study to the theory had to be discussed and cited properly.
Abstract:
The abstract too wordy and not very clear. Very long sentences (line 16 to 19).
“Inverted U shape” is constantly repeated in the text as well, what does “inverted U shape” correspond to in this context for the non-economist - line 20
No idea what is meant by "2)The trade comparative advantage of manufacturing industry is not conducive to industrial transformation, but under the constraints of environmental regulation, the inhibition effect of trade patterns on manufacturing transformation will transform into a promoting role."
Line 24- by "therefore, it is considered"; do you mean it is suggested/ we suggest? these are implications of the research findings.
Line 29-optimize the structure of import and export trade, meaning?
Introduction:
line 38- what is meant by environmental supervision? monitoring?
line 42- that environmental regulations were not only a crucial factor affecting the comparative advantage of a country's trade- negatively or positively? Can you expand on this ?
line 46- the dual characteristics of China's economic development? what do you mean by this ?
line 80- one type of research? did you mean line of research? could you please say what type/line of research it is between commas, after one type of research if it could be a certain type. Same comment for the line 95 (another kind of research).
line 67- what are the factors driving competitive advantage?
Line 111- what does "green industrial transformation and upgrading" mean?
Line 116- "adjustment of industrial structure" to what?
Line 128- "Research" is not a synonym for "research papers", and when used as a noun it is usually thought of as uncountable
Line 130-But there is little literature study- the word study here is redundant.
Line 135- clear about
Last paragraph in the introduction is very long, it has to be either reduced in word count or divided into 2-3seperate paragraphs, one definitely starting from the innovation points in line 142.
Author Response
Point 1: The objective of the paper in the writers' own words: to explore the impact of China's comparative advantage of trade on the green transformation and upgrading of manufacturing industry under the background of strengthened environmental regulations, and provide suggestions for China to formulate appropriate environmental regulations and trade policies to promote the green transformation and upgrading of manufacturing industry” which is investigated in the regression analysis. This is an important and relevant question considering the very significant share and role of China in the world economy and manufacturing. However, this high importance is not highlighted with some relevant figures in the text.
Response 1: In order to highlight the importance of the manuscript's subject, the author has modified both the introduction and the theoretical part. At the same time, some figures have been added to illustrate the current state of comparative advantage of China's manufacturing exports. Eg “World Bank data show that since 2003, China's manufacturing exports have accounted for more than 90% of merchandise exports. However, manufacturing is a highly polluting sector, which has led to a lot of energy consumption in China. Therefore, in order to eliminate the low-end lock-in status of the manufacturing industry, its transformation and upgrading are essential, and these processes depend on the implementation of regional environmental regulations to a certain extent”
Point 2: As it is clear from the sentence stating the manuscript’s objective above and as from the rest of the text, not much attention is paid to the use of language. The potential of the manuscript could much better be released, if it is revised and edited thoroughly before any resubmission. The review was difficult due to the wordy way the manuscript was written. Whole text must be in shorter, clearer sentences and with less repetition as it was in the parts on regression analysis and data (Sections 4 and 5). To illustrate my point I attach more particular comments by section at the end referring the line number. Off course, reading the whole text at this level of completion this critically would not be possible as a reviewer.
Response 2: We have used English editing service by MDPI. This manuscript has systematically modified the language.
Point 3: The literature review does not formulate the problem, its significance and the literature gap. It does not highlight the originality of the research. A brief Google scholar search showed me that there is a considerable body of recent literature on green industrial transformation in China. Citing these articles and discussing the contribution made with regards to gaps in this literature, justification of theory and quantitative methods used, (e.g. the advantage of methods use for analysis in the existing literature, why Cole et at.’ s method is used for environmental regulation strength measurement but not another method) would significantly improve the research formulation.
Response 3: This manuscript has systematically modified the literature review section, and explained why the Cole et al.’s method was used.
Point 4: There is a theory section but no linkage between theory and discussions. Discussions are not fully validated with the findings. In section 6, if there are any the theoretical and practical contributions of this study to the theory had to be discussed and cited properly.
Response 4: This manuscript has systematically modified the theory section. In section 6, the theoretical and practical contributions of this study to the theory has discussed and cited.
Point 5: The abstract too wordy and not very clear. Very long sentences (line 16 to 19).
Response 5: This manuscript has systematically modified the language.
Point 6: “Inverted U shape” is constantly repeated in the text as well, what does “inverted U shape” correspond to in this context for the non-economist - line 20.
Response 6: This manuscript has explained the inverted U-shaped curve in the text. Eg: “There is a significant inverted "U"-shaped curve relationship between environmental regulation and the transformation of the manufacturing industry. In other words, as environmental regulation increases, its impact on the transformation and upgrading of the manufacturing industry is first promoted and then suppressed.”
Point 7: No idea what is meant by "2)The trade comparative advantage of manufacturing industry is not conducive to industrial transformation, but under the constraints of environmental regulation, the inhibition effect of trade patterns on manufacturing transformation will transform into a promoting role."
Response 7: This manuscript has revised point 2. “When there are no environmental regulations, the trade comparative advantage of the manufacturing industry is not conducive to industrial transformation. However, under the constraints of environmental regulations, the comparative advantage of trade will significantly promote the green transformation and upgrading of manufacturing.”
Point 8: Line 24- by "therefore, it is considered"; do you mean it is suggested/ we suggest? these are implications of the research findings.
Response 8: This manuscript has made some changes to the language. “Therefore, in order to effectively promote the transformation and upgrading of manufacturing, this paper proposes the following policy recommendations:”
Point 9: Line 29-optimize the structure of import and export trade, meaning?
Response 9: It means diversifying the import and export of manufacturing and promoting its transformation to high-end manufacturing.
Point 10: line 38- what is meant by environmental supervision? monitoring?
Response 10: It means environmental regulations. The author has made changes.
Point 11: line 42- that environmental regulations were not only a crucial factor affecting the comparative advantage of a country's trade- negatively or positively? Can you expand on this ?
Response 11: The conclusions of the existing studies are not consistent, so the corresponding literature added by the author in this part is used as support.
Point 12: line 46- the dual characteristics of China's economic development? what do you mean by this ?
Response 12: It means the economic structure of modern industry and backward traditional agriculture.
Point 13: line 80- one type of research? did you mean line of research? could you please say what type/line of research it is between commas, after one type of research if it could be a certain type. Same comment for the line 95 (another kind of research).
Response 13: It means the opinions of different scholars, the author has modified the relevant language
Point 14: line 67- what are the factors driving competitive advantage?
Response 14: Considering the logical relationship between literature review, the author has deleted this paper
Point 15: Line 111- what does "green industrial transformation and upgrading" mean?
Response 15: It means green transformation and upgrading of the industry, the author has modified the language
Point 16: Line 116- "adjustment of industrial structure" to what?
Response 16: It means green transformation and upgrading of the industry, the author has modified the language
Point 17: Line 111- what does "green industrial transformation and upgrading" mean?
Response 17: It means increasing the proportion of technology-intensive and knowledge-intensive industries.
Point 18: Line 128- "Research" is not a synonym for "research papers", and when used as a noun it is usually thought of as uncountable
Response 18: The author has modified the sentence.
Point 19: Line 130-But there is little literature study- the word study here is redundant.
Response 19: The author has modified the language.
Point 20: clear about
Response 20: The author has modified the language. “we must clarify the impact mechanism of environmental regulation on the comparative advantage of industry trade”
Point 21: Last paragraph in the introduction is very long, it has to be either reduced in word count or divided into 2-3seperate paragraphs, one definitely starting from the innovation points in line 142.
Response 21: The author has divided
Reviewer 2 Report
I found the contents of theoretical parts (Section 2) a little poor, because most descriptions are elementary rather than academic and there are no clear explanations on the relationship between figures in this part and what the authors try to analyze (especially how the regulation affects GTFP). Moreover, Figures 2 and 3 are full of typos and hard to understand. So I would suggest that the authors should make Section 2 very concise by cutting unnecessary graphical explanations. Section 1 is also redundant and needs extensive editing.
As for empirical parts, I would like the authors to add more detailed explanations on how to derive GTFP which is the most important measure in this study and the definition of ML index in page 7.
I could not find any statements on comparison of the results of this study with those in other studies, so the authors should add some explanations.
Author Response
Point 1: I found the contents of theoretical parts (Section 2) a little poor, because most descriptions are elementary rather than academic and there are no clear explanations on the relationship between figures in this part and what the authors try to analyze (especially how the regulation affects GTFP). Moreover, Figures 2 and 3 are full of typos and hard to understand. So I would suggest that the authors should make Section 2 very concise by cutting unnecessary graphical explanations. Section 1 is also redundant and needs extensive editing.
Response 1: The author has modified the theoretical part. On the one hand, the diagram was deleted, and on the other hand, a more systematic explanation of the impact mechanism was made.
Point 2: As for empirical parts, I would like the authors to add more detailed explanations on how to derive GTFP which is the most important measure in this study and the definition of ML index in page 7.
Response 2: The author has explained the measurement of GTFP in detail according to the comments of reviewers and given the definition of ML index
Point 3: I could not find any statements on comparison of the results of this study with those in other studies, so the authors should add some explanations.
Response 3: The author has explained the research results in more detail, and added some literature for comparison.
Round 2
Reviewer 1 Report
Thanks for the corrections.
Reviewer 2 Report
I am satisfied with the revisions the authors have made.